

# Cross-cultural adaptation and validation of a Resistance Training Skill Battery for use in Chinese-speaking adolescents

Xiaolu Zha[1], David Lubans[2,3], Jordan Smith[3] and Ran Wang[4]

[1] School of Physical Education, Shanghai University of Sport, Shanghai, China
[2] Faculty of Sport and Health Sciences, University of Jyväskylä, Jyväskylä, Finland
[3] Centre for Active Living and Learning, University of Newcastle, Callaghan, New South Wales, Australia
[4] School of Athletic Performance, Shanghai University of Sport, Shanghai, China

## ABSTRACT

**Purpose**. Although the Resistance Training Skills Battery (RTSB) is a well-established instrument for assessing resistance training (RT) skill competency in Western populations, its applicability to Chinese adolescents remains unvalidated. This study aimed to translate, cross-culturally adapted, and validated the RTSB for Mandarin-speaking Chinese adolescents.

**Methods**. Employing established guidelines for cross-cultural adaptation, our study involved translating the RTSB into Chinese (RTSB-C), followed by back-translation, adaptation, and validation. The study quantified test-retest reliability, intra- and inter-rater reliability, and construct validity. Adolescents aged 12–16 years ($n = 64$) underwent two RTSB-C assessments, two weeks apart. Muscular fitness was evaluated using muscle fitness tests (handgrip strength, timed push-up, and countermovement jumps) to establish criterion validity.

**Results**. The RTSB-C demonstrated fair to excellent test-retest reliability across the two timepoints, the intra-class coefficient (ICC) for the RTSB-C was notably high at 0.94 (95% CI [0.90–0.95]); The intra-rater reliability was 0.94 ($p < 0.001$), and inter-rater reliability was 0.41–0.93 ($p < 0.05$). Construct validity was confirmed through linear regression analysis, with the model accounting for 46.8% of the variance in muscle fitness scores (MFS), and with gender ($r = 0.47$, $p < 0.001$) and RTSB-C scores ($r = 0.45$, $p < 0.001$) emerging as significant predictors.

**Conclusions**. We successfully translated and cross-culturally adapted the RTSB from English to Chinese, affirming its reliability and validity in assessing the RT skills of Chinese-speaking adolescents. The RTSB-C is recommended for accurate evaluation of RT competence in this demographic.

Corresponding author
Ran Wang, wangran@sus.edu.cn

# INTRODUCTION

Muscular strength plays a pivotal role in coordinating physical movements by mobilizing the skeletal system (*Goodway, Ozmun & Gallahue, 2019*) and optimizing movement efficiency. This is especially crucial during the dynamic periods of growth and maturation in childhood and adolescence, where adequate strength development not only mitigates

sports-related injuries but also enhances motor performance (*Faigenbaum & Myer, 2010*; *Myer et al., 2011*). Aligning with global health guidelines, the *World Health Organization (2020)* advocates for youth to engage in muscle-strengthening activities ≥3 times weekly, underscoring muscular strength as a determinant of motor competence (*Lloyd et al., 2014*).

Resistance training (RT), a specialized method of muscular conditioning aimed at enhancing health, fitness, and performance (*Faigenbaum et al., 2009*), utilizes various forms of resistance such as body weight, free weights, weight machines, elastic bands, and medicine balls. Robust evidence supports its safety and efficacy in youth when guided by qualified professionals (*Behm et al., 2008*; *Lloyd et al., 2014*; *Stricker et al., 2020*) making it a cornerstone of musculoskeletal development. Traditional RT assessments prioritize outcome metrics (*e.g.*, load intensity, repetitions), but neglected the process evaluation (*e.g.*, RT skills performance during the assessment). A process-oriented skill assessment involves identifying the 'presence' or 'absence' of several components/criteria per skill considered essential for mastering specific skills (*Barnett et al., 2015*). Failure to develop RT skills may lead to injuries and inadequate physical activity, as positive associations exist between motor skills and physical activity in children and adolescents (*Lubans et al., 2010*).

The Resistance Training Skills Battery (RTSB) is a process-oriented tool for assessing RT skill competence, that includes six foundational skills: bodyweight squat, push-up, lunge, suspended row, standing overhead press, and front support with chest touches. Skills are evaluated based on the successful demonstration of specific criteria, with scores reflecting the level of RT competency (*Lubans et al., 2014*). The RTSB has been validated for its construct validity, test-retest reliability, and intra/inter-rater reliability among adolescents (*Barnett et al., 2015*; *Lubans et al., 2014*), making it a valuable tool for assessing RT skills and improving exercise adherence within both school and community settings. However, the RTSB's exclusivity to English impedes its global adoption (*Ko, Rosen & Brown, 2015*; *Li et al., 2019*; *Nauck & Lohrer, 2011*). The RTSB's applicability to Chinese youth remains unestablished, creating a critical gap in cross-cultural research and practice. Our study aimed to bridge this gap by developing a linguistically and culturally adapted RTSB-C (Chinese version), ensuring accessibility for Mandarin speakers.

## MATERIALS & METHODS

### Cross-cultural adaptation

Initially, the RTSB was adapted for Chinese use following the guidelines recommended by the American Association of Orthopedic Surgeons (AAOS) (*Beaton, Bombardier & Guillemin, 1998*). The process involved six-steps: (1) initial translation, (2) synthesis of the translations, (3) back translation, (4) developing the pre-final version for field testing, (5) testing the pre-final version, and (6) finalizing the Chinese version of RTSB (RTSB-C). All details can be seen in Fig. 1. Notably, the term "suspended row" can be translated into "悬挂划船 (xuánguàhuáchuán)" and "悬吊划船 (XuándiàoHuáchuán)". However, due to the connotations of capital punishment associated with the term " 悬吊 (Xuándiào)" in Chinese cultural memory, the expert committee meeting decided to use the first expression. Pretesting with 20 adolescents indicated no further issues, the study assessed the reliability and validity of the adapted version.

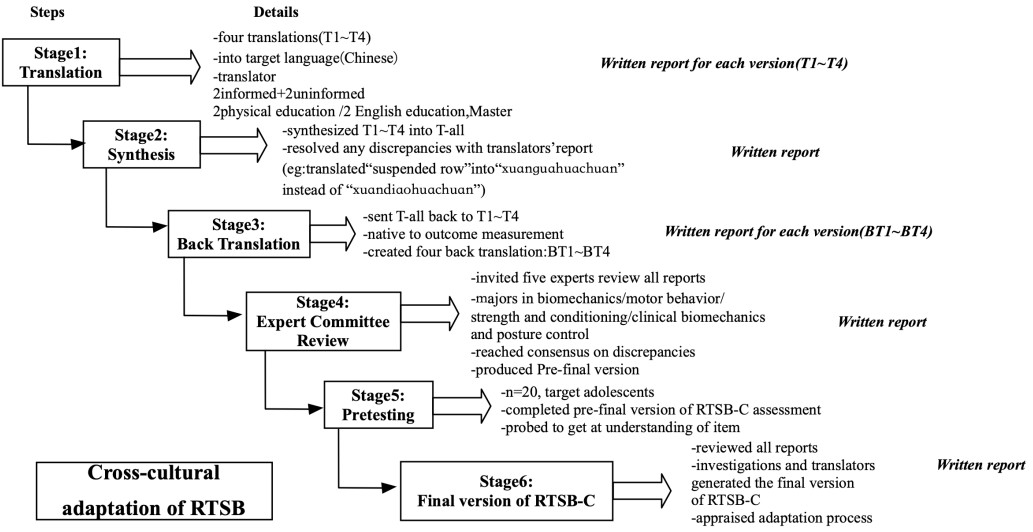

**Figure 1 Cross-cultural adaptation.** Note: T1, Translator1; BT1, Back Translator1.

## Participants

The study involved adolescents aged 12–16 years from Yixian middle school in Huangshan, Anhui province, China. The sample size was determined with reference to similar studies investigating resistance training and physical fitness in youth, which commonly enrolled between 50 to 70 participants (*Lubans et al., 2014*). All participants signed an informed consent form and verbal consent was obtained from their parents. Participants were ineligible if they had a medical conditions or physical injuries that would prevent them from taking part in the study, as reported by parents and participants at the start of the study. The Institutional Review Board of the Shanghai University of Sport approved this study (ethical approval ID 102772019RT013).

## Test procedures

Trained research assistants conducted two assessment sessions separated by one week to minimize learning effects and maturation variability, all research assistants participated in a full-day training workshop in preparation for the assessments (*Lubans et al., 2014*). Participants, divided by sex into groups of 5–6, completed the RTSB-C and muscle fitness tests (MFT) during the first session (Trial 1, hereafter called T1) and another RTSB-C test in the second (Trial 2, hereafter called T2). After a 10-minute standardized warm-up, which included jogging, dynamic stretches, the MFT test was performed. The sequence of RTSB-C was: (a) body weight squat, (b) push-up, (c) lunge, (d) suspended row, (e) standing overhead press, and (f) front support with chest touches. There were five scoring criteria for body weight squat, lunge, standing overhead press, and front support with chest touches, each scoring one point and requiring two repetitions, with a scoring range of 0–10 points; and there were four scoring criteria for push-up and suspended row, each scoring one point and requiring two repetitions, with a scoring range of 0–8 points, for a total of 28 scoring items, each scoring one point and requiring two repetitions, with a full

score of 56 points. The movement needs to be repeated twice out of a total of 56 points. Upon completion of all assessments, the video data from the RTSB-C were evaluated. A stratified random sampling approach was used to select the test videos of one-third of the participants for independent assessment by six raters, with the aim of calculating inter-rater reliability. Furthermore, one of the raters evaluated the same test video on two separate occasions to ascertain intra-rater reliability.

*RTSB-C.* The RTSB-C involved participants performing each skill twice under supervision after observed demonstrations by a trained assistant, with performances recorded for later analysis. A digital video camera was used to record all skills in accordance with standardised procedures (*Lubans et al., 2014*), with the camera positioned at a 45°-degree angle to the subject to enable front- and side-on views. This was considered to be the optimal viewing position to allow the raters to correctly assess each skill during the video analysis process (*Lubans et al., 2014*). Since all exercises only involved body weight or used a bar without additional weight, there was no prescribed standardized warm-up routine before participants engaged in the RTSB-C. Throughout the process, research assistants offered general encouragement to the participants, such as 'watch carefully', 'finish carefully', 'go for it', *etc.*, but without feedback specific to their skills. Six raters with backgrounds in physical education or sports science evaluated the recorded performances against RTSB-C criteria, determining a resistance training skill quotient (RTSQ) for each participant. The raters had varying experience in movement skill assessment and resistance training and completed a two-hour session covering RTSB theory and practice. It mainly focuses on the origin, composition, and evaluation rules of RTSB-C, and provides pretesting videos for raters to score on the spot.

*Muscle fitness tests (MFT).* Handgrip strength, timed push-ups, and countermovement jumps were selected as the assessment tests. This selection was based on their established reliability and validity for comparing upper- and lower-body muscle strength measures in adolescent populations (*Lubans et al., 2011*; *Ortega et al., 2008*).

Handgrip strength was measured using a dynamometer (FAB12-0604; JAMAR, Duluth, MN, USA). The spacing between the inner and outer stirrups of the grip gauge was customized to ensure optimal comfort based on the individual's palm circumference. When initiating the test, the participants maintained maximum force for two to three seconds. The measurement procedure was replicated three times for each hand, and the resultant values for both the left and right hands were averaged to derive a single kilogram value for statistical analysis.

The timed push-up test began with participants in the push-up position, hands and toes touching the floor, arms approximately shoulder width apart, and back straight. A metronome was set at 40 b min$^{-1}$, used to control the tempo of the push-ups to be one repetition per three seconds (*Lubans et al., 2014*). Participants lowered themselves to the floor with a 90° elbow flexion in a controlled manner and then pushed back up. The test was terminated when participant failed to lower their body to the required depth for three non-consecutive repetitions (with assistant verbal warning), failed to maintain the beat with metronome, or on voluntarily terminated. The total number of push-ups was recorded for statistical analysis.

Countermovement jumps began with participants standing on the force platforms (PASCO PS-2141) with hands on hips. Participants then rapidly squated down and jump as high as possible. The force-time data was recorded and processed through customized scripts (MATLAB, MathWorks, Inc., USA). The test was performed twice separated by a rest period of 10 s. The mean of the two jump heights was calculated for statistical analysis.

All test results were converted to standardized values (individual value minus group mean and divided by the standard deviation of the group) and summed up to calculate a Muscle Fitness Score (MFS) for statistical analysis.

### Data analysis

Data were analyzed using SPSS (version 22; IBM, Armonk, NY, USA). Descriptive statistics were computed as the mean and standard deviation. An ICC was employed to assess test-retest reliability, intra-rater reliability, and inter-rater reliability. The ICC values were interpreted as follows: with ICC > 0.75 indicating excellent reliability, 0.75−0.40 fair, and < 0.40 poor (*Fleiss, 1986*). Significance was set at $p < 0.05$. The construct validity was evaluated using multiple regression modeling, with gender, age, and RTSQ as independent variables and MFS as the dependent variable. Age exhibited no significant association with MFS and was therefore removed from the final model.

## RESULTS

### Study participants

A total of 64 adolescents (mean age = 13.53 ± 1.04 years, height = 1.58 ± 0.09 m, weight = 50.30 ± 10.60 kg), comprising 37 boys and 27 girls, ultimately participated in the study. According to China's BMI classification criteria (*Chengye, 2004*) for overweight and obesity screening in school-age children and adolescents, the majority of participants were classified as healthy, 28 boys (18.6 ± 1.5 kg m$^2$) and 27 girls (19.5 ± 2.1 kg m$^2$). However, there were instances of overweight (five boys, 23.2 ± 1.07 kg m$^2$) and obesity (four boys, 27.85 ± 2.71 kg m$^2$), as detailed in Table 1. The specific score distribution for each movement is detailed in Table 2. The scores for the six movements are predominantly concentrated between four to eight points. The push up received the lowest scores (with a mean of 4.80 points), while the standing overhead press had the highest scores (with a mean of 7.50 points). Most movements showed an improvement in scores from the first session (except for the standing overhead press), with minimal differences between males and females, and females generally performed better than males in the test results.

### Reliability

The ICC estimates and changes in mean are reported in Table 3. The ICC values for individual skills ranged from 0.61 (95% CI [0.43–0.77]) for the standing overhead press to 0.88 (95% CI [0.81–0.93]) for the lunge. The RTSQ demonstrated high stability with an ICC of 0.94 (95% CI [0.90–0.95]).

Intra-rater reliability was notably high with a correlation coefficient of 0.94 ($p < 0.001$). Inter-rater reliability was generally strong among five of the six raters ($p < 0.05$), with all ten pairwise correlations above 0.65, six above 0.70, and ranging from 0.65 to 0.93.

**Table 1** Characteristics of study participants ($N = 64$).

| Characteristics | Boys ($N = 37$) | | Girls ($N = 27$) | | Total ($N = 64$) | |
|---|---|---|---|---|---|---|
| | Mean | SD | Mean | SD | Mean | SD |
| Age (y) | 13.38 | 0.92 | 13.74 | 1.16 | 13.53 | 1.04 |
| Body mass (kg) | 53.28 | 12.01 | 46.42 | 6.65 | 50.39 | 10.60 |
| Height (m) | 161 | 0.10 | 154 | 0.05 | 1.58 | 0.09 |
| BMI (kg m$^{-2}$) | 20.46 | 3.44 | 19.54 | 2.07 | 20.07 | 2.96 |
| BMI category | | | / | | | |
| Healthy weight, $n$ (%) | 28 | 75.7 | 27 | 100 | 55 | 85.9 |
| Overweight, $n$ (%) | 5 | 13.5 | 0 | 0 | 5 | 7.8 |
| Obese, $n$ (%) | 4 | 10.8 | 0 | 0 | 4 | 6.3 |
| Handgrip strength test (kg) | 29.98 | 7.97 | 23.80 | 4.70 | 27.37 | 7.41 |
| Timed push-up test | 13.03 | 4.25 | 8.59 | 5.46 | 11.16 | 5.24 |
| Countermovement jump (cm) | 27.62 | 3.96 | 24.98 | 2.71 | 26.51 | 3.71 |

**Notes.**
BMI, body mass index.

However, the sixth rater's correlations with others were lower ($0.41-0.54$), and this rater also assigned the highest mean scores (41.61), as presented in Table 4.

## Construct validity

The final regression model accounted for 46.8% of the variance in MFS ($p < 0.001$), with both gender ($r = 0.47$, $p < 0.001$) and RTSQ ($r = 0.45$, $p < 0.001$) significantly predicting MFS. This suggests that higher resistance training skill competency is associated with better performance on muscular fitness tests.

## DISCUSSION

The significance of muscular fitness in adolescence, coupled with the rising interest in youth resistance training programs, underscores the necessity for a reliable tool to assess adolescents' resistance training skill competency. The RTSB is the first process-oriented tool designed for this purpose, enabling both individual and group-level performance assessments in resistance training programs. It could be used to assess each participant's performance and, when appropriate, provide specific information regarding group-level performance and progress in youth resistance training programs while providing constructive feedback to participants (*Barnett et al., 2015*). As each criterion is quantifiable, subsequent exercises can be targeted if participants do not meet the point. Recognizing the absence of a validated instrument for assessing resistance training skills in China, our study sought to fill this gap by developing a Chinese version of the RTSB. Following the guidelines by *Beaton, Bombardier & Guillemin (1998)*, we tailored the tool for Chinese-speaking adolescents through meticulous translation and cross-cultural adaptation of the Australian RTSB, which is internationally recognized, efficient, and brief.

The reliability and construct validity of the RTSB have been well-documented in prior research, confirming its efficacy in evaluating adolescents' resistance training skill competency (*Barnett et al., 2015*; *Lubans et al., 2014*). Our study extended this validation

Zha et al. (2025), *PeerJ*, DOI 10.7717/peerj.20387

**Table 2** Results from RTSB-C in boys (N = 37) and girls (N = 27) (mean ± SD).

| RTSB-C | Range | T1 | | | T2 | | | T2-T1 | | |
|---|---|---|---|---|---|---|---|---|---|---|
| | | **All** | **Boys** | **Girls** | **All** | **Boys** | **Girls** | **All** | **Boys** | **Girls** |
| Body weight squat | 3–10 | 6.64 (1.97) | 6.41 (2.11) | 6.96 (1.74) | 6.84 (1.77) | 6.32 (1.76) | 7.56 (1.55) | 0.20 (−0.20) | −0.67 (−0.03) | 1.30 (−0.82) |
| Push up | 2–8 | 4.80 (2.08) | 4.84 (2.29) | 4.74 (1.79) | 5.36 (1.78) | 5.32 (1.86) | 5.41 (1.69) | 0.56 (−0.30) | 0.64 (0.07) | 0.75 (−0.74) |
| Lunge | 2–10 | 6.83 (2.27) | 6.43 (2.35) | 7.37 (2.08) | 7.17 (1.97) | 6.95 (2.09) | 7.48 (1.78) | 0.34 (−0.30) | −0.29 (−0.20) | 1.12 (−0.61) |
| Suspended row | 1–8 | 5.34 (1.90) | 4.89 (1.97) | 5.96 (1.63) | 5.86 (1.74) | 5.32 (1.89) | 6.59 (1.22) | 0.52 (−0.15) | −0.61 (0.25) | 1.74 (−0.79) |
| Standing overhead press | 4–10 | 7.50 (1.50) | 7.08 (1.59) | 8.07 (1.17) | 7.41 (1.49) | 7.35 (1.48) | 7.48 (1.53) | −0.09 (−0.01) | −0.75 (0.24) | 0.40 (−0.06) |
| Front support with chest touch | 2–10 | 6.31 (2.53) | 5.92 (2.64) | 6.85 (2.32) | 6.66 (1.99) | 6.38 (2.02) | 7.04 (1.93) | 0.34 (−0.54) | −0.18 (−0.40) | 1.08 (−0.82) |
| RTSQ | 14–56 | 37.42 (7.69) | 35.57 (8.28) | 39.96 (6.07) | 39.14 (6.34) | 37.38 (7.01) | 41.96 (4.33) | 1.72 (−1.36) | −1.10 (0.59) | 2.40 (−4.30) |

**Notes.**

RTSQ, resistance training skill quotient; T1, Trial 1; T2, Trial 2.

**Table 3  Reliability of RTSB-C.**

| RTSQ-C | ICC (95% CI) | Change in mean | |
| --- | --- | --- | --- |
| | | Mean (SD) | $p$ |
| Body weight squat | 0.85 (0.78–0.90) | 0.20 (−0.20) | 0.124 |
| Push-up | 0.85 (0.79–0.91) | 0.56 (−0.30) | 0.000[**] |
| Lunge | 0.88 (0.81–0.93) | 0.34 (−0.30) | 0.013[*] |
| Suspended row | 0.85 (0.80–0.91) | 0.52 (−0.15) | 0.007[**] |
| Standing overhead press | 0.61 (0.43–0.77) | −0.09 (−0.01) | 0.575 |
| Front support with chest touch | 0.86 (0.75–0.92) | 0.34 (−0.54) | 0.035[*] |
| RTSQ | 0.94 (0.90–0.95) | 1.72 (−1.36) | 0.000[**] |

Notes.

RTSQ, resistance training skill quotient.
[*]$p < 0.05$.
[**]$p < 0.01$.

**Table 4  Correlations between raters and summary statistics for each rater.**

| Raters | Inter-rater reliability | | | | | | Mean | Median | Max | Min |
| --- | --- | --- | --- | --- | --- | --- | --- | --- | --- | --- |
| Rater 6 | 1 | | | | | | 41.61 | 41.0 | 53 | 27 |
| Rater 5 | 0.42 | 1 | | | | | 37.63 | 38.0 | 54 | 23 |
| Rater 4 | 0.44 | 0.93 | 1 | | | | 40.44 | 41.0 | 52 | 26 |
| Rater 3 | 0.51 | 0.65 | 0.81 | 1 | | | 39.14 | 39.5 | 53 | 21 |
| Rater 2 | 0.54 | 0.66 | 0.74 | 0.69 | 1 | | 34.92 | 34.0 | 51 | 22 |
| Rater 1 | 0.41 | 0.70 | 0.76 | 0.68 | 0.93 | 1 | 35.56 | 35.0 | 45 | 21 |
| | R6 | R5 | R4 | R3 | R2 | R1 | 38.22 | 38.2 | 51 | 23 |

to a Chinese-speaking sample through the RTSB-C, revealing that the adapted version was comprehensible and yielded robust statistical outcomes. Notably, the test-retest reliability for the RTSB-C (ICC = 0.94) surpassed that of the original English version (ICC = 0.88), as reported by *Lubans et al. (2014)*. Among the raters, five demonstrated a high level of agreement (0.65−0.93), while the sixth rater's agreement was notably lower (0.41−0.55), underscoring a significant variance in rater consistency. The importance of rater qualifications has been emphasized in previous studies employing process-oriented assessments of children's movement skills, with raters typically designated as "experts" due to their substantial expertise and background in assessing movement skills (*Myer et al., 2011*). However, not all raters in our study met the conventional "expert" criteria, as evidenced by the varied experience levels and the resultant disparities in scoring outcomes. Intriguingly, the sixth rater, because of limited prior experience in skill assessment, assigned the highest mean score. This observation suggests that prior experience in evaluating movement skills and resistance training could contribute to more consistent and accurate assessments in future research using the RTSB-C. Therefore, we recommend that future applications of the RTSB-C clearly report the rater's background in movement skill assessment. This transparency is crucial for interpreting potential rater-induced variability and will enhance the comparability and reliability of findings across different research settings.

The final regression model of our study accounted for 46.8% of the variance in MFS, a substantial increase compared to the 39% variance explained in the original study by *Lubans et al. (2014)*. This improvement in explanatory power may be attributed to our methodological modification: substituting the long jump with the countermovement jump. The countermovement jump is recognized for its higher reliability and validity in assessing the explosive power of the lower limbs, as noted by *Ortega et al. (2008)*. Despite its advantages, the requirement for force platforms to perform the countermovement jump introduces a level of complexity not present in the simpler standing long jump, potentially restricting its application in school environments due to logistical and equipment constraints. Our findings lend support to the hypothesis that replacing the long jump with the countermovement jump could yield more accurate assessments of muscular fitness, as evidenced by the enhanced validity of our regression model compared to that of the original RTSB version. This adjustment underscores the balance between methodological rigor and practicality in the context of physical education settings, highlighting the need for further innovation in developing accessible yet accurate measures of muscular fitness in youth.

Recent findings highlight the unique and additional benefits of muscle-strengthening activities, or resistance training, for children and adolescents (*Robinson et al., 2023*; *Smith et al., 2020*). Surveillance data reveal a wide range of participation rates in resistance training globally, from a minimal 0.3% to as high as 12.4% among adolescents (*Hulteen et al., 2017*). This variability can partly be attributed to prevailing concerns among parents, many of whom view resistance training as inappropriate or risky for young people (*Hoor et al., 2015*). Moreover, the focus of physical education (PE) programs is often on traditional team sports and their related motor skills, which further limits opportunities for specialized resistance training instruction (*Ennis, 2014*). These perspectives contribute to a significant gap in exposure to and education about resistance training, underscoring the need for enhanced awareness and incorporation of such activities in youth fitness programs. The teacher's role as a trusted authority figure in Chinese school provided a powerful channel for disseminating new pedagogical approaches. By equipping physical education teachers with the knowledge and tools to effectively assess and teach foundational resistance training skills, their authority can become a primary driver for promoting resistance training in the school curriculum. This approach aligns the introduction of a new assessment tool with established cultural norms, making its adoption more likely to be successful and sustainable.

Our research confirms a significant positive correlation between muscular fitness and resistance training skill competency among both male and female adolescents, echoing conclusions drawn by previous studies (*Pichardo et al., 2019*; *Smith et al., 2018*). This correlation underscores the importance of establishing foundational resistance training competencies at an early age. We advocate for ensuring that children and adolescents possess a solid grasp of these essential skills prior to embarking on resistance training programs involving additional external resistance (*Lloyd et al., 2014*). Mastery of technical aspects should be considered as critical as understanding the fundamental principles of resistance training itself. Therefore, before progressing to more structured and advanced

resistance training, adolescents should exhibit proficiency in resistance training skills, laying a groundwork for safe and effective practice. This approach not only enhances the safety and efficacy of resistance training but also supports long-term fitness and health outcomes.

While our study successfully adapted the RTSB for Chinese adolescents and validated its reliability and validity, several limitations must be acknowledged. First, the intra-rater reliability was indeed assessed by a single experienced rater. We fully acknowledge that relying on one rater may restrict generalizability of the results, as the variations in scoring that are common in movement quality assessments were not accounted for. Second, we did not explore the influence of gender (*Smith et al., 2018*), socioeconomic status, or BMI-based health status on RTSB-C performance. Third, the absence of longitudinal data limits our understanding of how RTSB-C scores might correlate with changes in health outcomes over time. Finally, although schools play a critical role in fostering RT skill development among youth, suggesting that the RTSB-C could serve as both a diagnostic tool to gauge readiness for advanced resistance training and a means to assess educational outcomes in physical education. However, the potential of RTSB-C to inform program development and evaluate student progress over time remains underexplored. Future research should aim to bridge these gaps by investigating the relationship between RT skills and health outcomes more thoroughly, incorporating longitudinal and experimental designs to clarify causality and the impact of RTSB-C score changes on health. Expanding the study to include diverse regions and ethnic groups within the Chinese adolescent population would also enhance the generalizability of our findings.

## CONCLUSIONS

Our study marks a significant step achieving international comparability in the field of resistance training skills research, as it has involved the development and validation of a Chinese version of the RTSB (RTSB-C). This cross-culturally adapted tool has demonstrated high reliability and validity, making it suitable for evaluating the resistance training competencies of Chinese-speaking adolescents. Such evaluations can provide valuable benchmarks for assessing individual and group performance in resistance training programs, both in educational and community settings. The application of the RTSB-C extends beyond academic interest, offering practical benefits for PE teachers and program developers by facilitating the assessment of student progress and the effectiveness of resistance training interventions. By establishing a foundation for cross-national comparisons of RT skills, this study paves the way for further research that could enhance our understanding of the role of resistance training in adolescent development and health. Future efforts should continue to expand the evidence base by exploring the utility of the RTSB-C across different cultural contexts and its implications for promoting motor skill development and health outcomes in adolescents.

## ACKNOWLEDGEMENTS

The authors would like to thank the following content experts for providing feedback on the Chinese version of the RTSB: Dr. Boyi Dai, Dr. Qin Zhu, and Dr. Yumeng Li. The authors would also like to thank the participants and their parents for their participation in the study.

### Funding

This study was supported by the Shanghai Key Lab of Human Performance (Shanghai University of sport) (NO. 11DZ2261100). The funders had no role in study design, data collection and analysis, decision to publish, or preparation of the manuscript.

### Grant Disclosures

The following grant information was disclosed by the authors:
Shanghai Key Lab of Human Performance (Shanghai University of sport): NO. 11DZ2261100.

### Competing Interests

The authors declare there are no competing interests.

### Author Contributions

- Xiaolu Zha conceived and designed the experiments, performed the experiments, analyzed the data, prepared figures and/or tables, authored or reviewed drafts of the article, and approved the final draft.
- David Lubans conceived and designed the experiments, authored or reviewed drafts of the article, and approved the final draft.
- Jordan Smith conceived and designed the experiments, authored or reviewed drafts of the article, and approved the final draft.
- Ran Wang conceived and designed the experiments, performed the experiments, analyzed the data, authored or reviewed drafts of the article, and approved the final draft.

### Human Ethics

The following information was supplied relating to ethical approvals (*i.e.*, approving body and any reference numbers):

Shanghai University of Sport granted Ethical approval to carry out the study within its facilities.

### Data Availability

The raw measurements are available in the Supplemental File.

## Supplemental Information

Supplemental information for this article can be found online at http://dx.doi.org/10.7717/peerj.20387#supplemental-information.

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
