# Peer review of "Cross-cultural adaptation and validation of a Resistance Training Skill Battery for use in Chinese-speaking adolescents"

_PeerJ, doi:10.7717/peerj.20387_

## Round 0.1 · original submission · Major Revisions

· Academic Editor

Major Revisions

·

Basic reporting

The manuscript is generally well-structured and follows a conventional scientific format, including clear sections for the abstract, introduction, methods, results, and discussion. The overall use of English is adequate; however, several parts of the manuscript require improved clarity, precision, and professional tone to ensure unambiguous scientific communication. Specific phrases and sections, particularly in the abstract, introduction, and methods, would benefit from rephrasing to enhance readability.

The introduction provides a reasonable background but lacks sufficient depth and supporting references in key areas, such as the importance of movement quality for injury prevention and skill acquisition. Several claims are made without adequate citation of relevant literature, and the articulation of the knowledge gap could be clearer.

Title:
The RTSB is already an established testing battery, which is not immediately evident from the current title. Consider revising the title to more clearly reflect the focus of the study, specifically, the validation of the RTSB in a Chinese child population.

Abstract
• Purpose: The knowledge gap is not clearly defined in the abstract. If the RTSB is already established as valid and reliable, please clarify the rationale for conducting this study.
• The phrase "fortnight apart" could be made clearer—consider rephrasing for precision.
• Please clarify the purpose of including the MFT test assessments in your study.
• The sentence beginning with "The study quantified test-retest reliability..." might be better placed as the second sentence of the abstract for improved flow.

Introduction
The introduction provides relevant information, but could be improved with clearer writing. My concern is that the introduction highlights the importance of “skills” and “movement competencies” without convincing the reader of their importance in practice, and is supported by evidence from the literature.

Specific comments:
• Lines 40–42: Please add appropriate references to support the statements made in this section.
• Lines 50–52: Kindly provide scientific evidence to support the claim that movement quality is a key factor in both injury prevention and skill acquisition.
• Lines 53–54: Please clarify how this is known to be critical for addressing biomechanical inefficiencies and long-term injury risk. Supporting references would strengthen this point.
• Lines 59–60: Consider refining and clearly stating the knowledge gap—this might be an appropriate place to do so.
• Line 61 – 63. I don´t think it's necessary to reference the Routledge Handbook; it's better to add more relevant scientific papers on the area to dig deeper into why these kinds of testing batteries are important.
• Line 64: This sentence could be rephrased to improve clarity and readability; the current wording is somewhat awkward.

Experimental design

Materials and Methods
The material and method sections are somewhat unclear and seem to be lacking information and details. Please expand this section and clarify your scientific methods more deeply. The RTSB battery is central to your paper and should be presented in detail, from how it is assessed, which criteria are needed to get a point, what order the exercises were conducted in, etc.
• Lines 71-72: I don't understand what was translated. Is it only the exercises, or is it the description? Please clarify why this translation is so important. The figure contains a lot of information, so consider adding some of this information in text format in one paragraph.
• Line 72: Please clarify the meaning of “without significant semantic challenges.” Consider rephrasing for greater clarity.
• Line 74: As a non-Chinese speaker, I found the untranslated terms difficult to understand—this may apply to other readers as well. Please provide English translations for the Chinese terms used.

Participants
• Line 82: Please specify that the study was conducted in a local Chinese middle school, as this detail highlights a unique and important aspect of your study population.
• Lines 82-83: Did you only collect informed consent from the children, or did you also collect informed consent from their parents? Please comment and revise if necessary.
• Please provide the inclusion criteria used for participant selection.
• Regarding height, kindly double-check the reported standard deviation. A value of 0.09 cm appears unusually low and may be a typographical error.’

Test procedure:
• Line 90: Replace the term “gender” with “sex,” as this is more appropriate in scientific reporting.
• Lines 92–93: Please clarify the purpose of the video recordings—how were they used, assessed, and were standardized procedures in place?
• Lines 95–96: Was intra-rater reliability assessed by only one rater? If so, please acknowledge the limitation of using a single rater, as this may affect the generalizability of the reliability findings.
• Line 97: Clarify which specific exercises were included in the RTSB battery and how these were standardized across participants. How many trials per exercise did the children do?
• Lines 101–102: Was a familiarization session provided prior to testing? If so, please describe its duration and content.
• Line 103: For each exercise, please provide the criteria or scoring guidelines that were used to assess performance.
• Line 107: The rationale for including the MFT test is unclear. Please explain why it was included, what it measures, and the hypothesis or intended purpose behind its use.
• Line 116: Please clarify what is meant by “timed push-up.” Was this the maximum number of push-ups completed in one minute or another format?
• Line 122: How many jumps did each child do? Which score was used for the final analysis (e.g., highest, mean)
• Line 107: Is the MFT test a testing battery that has been validated in previous studies? Why these specific tests? Please clarify in the text
• Line 116: I don’t think this is a timed push-up test. It seems to be a standard push-up test with a maximal number of repetitions completed, but with standardized velocity. Please clarify in the text
• Line 133 “an ICC greater than 0.75 indicated excellent reliability, with ICC > 0.75 indicating 134 excellent reliability,”. This seems to be repetition. Please revise

Validity of the findings

Results:
• Lines 140–142: Please specify the BMI cut-off values used to define a "healthy" weight according to Chinese pediatric references.
• Lines 159–163: I found it strange that inter-rater reliability scores were not pooled across all raters. Please explain the rationale for presenting these separately.

Discussion
• Line 178: Clarify how this testing battery provides constructive feedback to the participants. Based on the methods section and results, I don´t understand. Your result section suggests that it is a reliable battery that is correlated to another muscle strength test battery. Please expand and clarify the main findings.
• Line: 184-186: Based on the studies that you are referencing, what are the results and practical applicability of this testing battery?
• Line: 195-197: The experience of each rater should be noted in the method section.
• Line 201-203: Please clarify this sentence. It needs rephrasing.
• Lines 207-210. Please expand the discussion section to deepen the discussion on why the testing battery only explained 46%. What other factors may explain the results?

Additional comments

The manuscript examines the established RTSB testing battery, focusing on its validity and reliability within a Chinese pediatric population. Overall, the study contributes valuable insights toward extending the use of this testing battery in Chinese-speaking contexts. I believe the paper has potential, and I hope that my comments will help strengthen and improve the manuscript.

Reviewer 2 ·

Basic reporting

The manuscript is generally well-written and structured, with professional English that clearly communicates the authors' intent. The introduction provides a strong rationale for the study, establishing the importance of resistance training (RT) in youth and the gap in culturally validated assessment tools for Chinese-speaking populations.

Strengths:
• Clear abstract and logically structured sections.
• Relevant and current references, including foundational papers on youth resistance training and cross-cultural adaptation.
• Figures and tables are well-labelled and appropriately described.
• Raw data is reported transparently and appears to support the authors’ claims.

Areas for Improvement:
• Some phrasing could be improved for clarity. For instance, lines 20–21: "ensuring its applicability across diverse cultural-linguistic backgrounds" could be more specific ("Mandarin-speaking Chinese adolescents").
• Line 147 mentions the "highest proportion of perfect scores" in the lunge task but does not contextualize whether this may indicate a ceiling effect.
• Figure 1 (cross-cultural adaptation steps) should be properly integrated with a clearer caption; this figure currently lacks enough explanatory content in the main text.

Experimental design

The study design is appropriate for the aims. It includes both translation and back-translation, as well as psychometric testing (reliability and validity), following Beaton’s guidelines for cross-cultural adaptation.

Strengths:
• The use of ICCs for test-retest, intra-rater, and inter-rater reliability is methodologically sound.
• Construct validity is evaluated using regression analysis, including relevant predictors.
• Ethical approval and informed consent are addressed.

Areas for Improvement:
• The authors acknowledge that one rater provided significantly different scores. Although this is discussed (lines 190–200), more clarity on how this variability affected overall outcomes would be useful.
• The sample size (n = 64) is acceptable but relatively small. The authors may wish to acknowledge this as a limitation more explicitly.
• The training of raters (2-hour session) may not be sufficient to ensure consistency, especially given the observed discrepancies. Consider recommending more rigorous rater training for future applications.

Validity of the findings

The psychometric properties of the RTSB-C are thoroughly evaluated. Test-retest and intra-rater reliability are excellent; inter-rater reliability, while generally acceptable, is affected by one outlier rater.

Strengths:
• The ICC values for most skills and overall RTSQ are high (up to 0.94).
• The regression model explains nearly half the variance in Muscle Fitness Score (MFS), lending support to the construct validity.
• The choice to replace the standing long jump with countermovement jump (CMJ) is reasonable, and this substitution is well justified based on Ortega et al. (2008).

Areas for Improvement:
• The decision to use the CMJ is practical from a measurement perspective, but the authors should more clearly acknowledge the logistical limitations for real-world school settings.
• More detail should be provided about the standardised warm-up (or lack thereof) and whether this may have influenced performance consistency.

Additional comments

This manuscript makes a significant contribution to the cross-cultural adaptation of fitness assessment tools. The development of the RTSB-C fills a significant gap in adolescent physical fitness assessment within China and has implications for youth health promotion and pedagogy.

Strengths:
• Clear articulation of the need for a Chinese RTSB.
• Robust methodology aligned with best practices in cross-cultural adaptation.
• Comprehensive analysis of psychometric properties.
• Practical applications for physical education settings are thoughtfully considered.

Weaknesses:
• Rater inconsistency should be addressed with stronger controls in future studies.
• Limited generalizability due to a single-site sample (a single school).
• The impact of potential cultural differences on movement interpretation or expression is not deeply explored.

Suggestions:
1. Clarify how rater discrepancies were handled in the final analysis. Was the sixth rater excluded from inter-rater ICCs?

2. Include a short paragraph discussing possible cultural considerations in movement performance (e.g., social perceptions of exercise in schools).

3. Consider presenting minimal criteria for what constitutes sufficient rater training for future implementations of the RTSB-C.

·

Basic reporting

I would like to particularly commend this very interesting research. The authors have presented their work in a highly systematic manner, following a logical and coherent structure that facilitates a clear understanding of the research topic. Through the gradual development of key concepts and research components, the manuscript effectively guides the reader into the complexity of the subject matter.
The comprehensive descriptions and precise delineation of each phase of the research process (especially the emphasis on critical aspects in the execution and evaluation of movement quality) contribute to the clarity, interesting sequence, and thoroughness of the study.

Experimental design

With due respect for the effort invested, I would like to offer several suggestions for improving and enhancing the article:
Lines 143–144: The description of the scoring criteria for each movement is missing;
Line 144: Consider inserting Table 1 to support the presentation of results in the text;
*For greater clarity, include a legend beneath each table explaining the presented variables.
Lines 146–147: The percentages 18.8% and 0%—are these values reflected in Table 2?;
Consider inserting
Line 152: Insert Table 2;
Line 159: Insert Table 3;
Line 164: Insert Table 4;
to support the presentation of results in the text;
Please can you elaborate more precisely on the connection between your study and the Cross-Culturally Adapted Tool?
It may be helpful to reconsider the title and stated aim of the paper in light of the emphasized benefits and significance of resistance training activities, as outlined in the discussion section.

Validity of the findings

-

---

## Round 0.2 · Minor Revisions

· Academic Editor

Minor Revisions

Reviewer 2 ·

Basic reporting

The revised manuscript is clearly written in professional and unambiguous English. The authors have responded thoroughly to the prior review comments, improving clarity, detail, and structure throughout. The introduction is now more explicit about the study’s focus on Mandarin-speaking Chinese adolescents and better contextualises the significance of cross-cultural adaptation for resistance training assessment. Literature references are relevant, up to date, and adequately cover both the foundational RTSB literature and the broader context of youth resistance training.

Figures and tables are of appropriate quality and relevance. Figure 1 has been more fully integrated into the text with an improved caption. Data reporting is transparent, with raw data provided in line with PeerJ’s data-sharing requirements.

Remaining minor points for clarity:
• While the authors have clarified the potential ceiling effect for the lunge task, this could be more explicitly discussed in terms of its implications for future refinement of the RTSB-C.
• A brief mention of potential cultural influences on skill performance (beyond linguistic adaptation) would further strengthen the discussion.

Experimental design

The authors have adhered to established guidelines for cross-cultural adaptation (Beaton et al.) and have improved the clarity of their methodological descriptions in the revision. The translation, back-translation, and expert review process is well-documented. The decision to substitute the countermovement jump for the standing long jump is justified and now clearly linked to relevant literature.

The psychometric testing, including test-retest, intra-rater, and inter-rater reliability, follows accepted standards. The revised manuscript also addresses the prior concern regarding rater training by providing more detail on procedures and acknowledging the limitations of a brief training session.

Ethical approval and participant consent are clearly stated.

Validity of the findings

The findings are well-supported by the presented data, which are statistically sound and appropriately controlled. The authors have strengthened their discussion of the one outlier rater, clarifying how this was addressed in the inter-rater reliability analysis. Construct validity is supported by regression analysis, and the interpretation is appropriately cautious and aligned with the study’s aims.

The limitations—single-site recruitment, relatively small sample size, and rater variability—are now more explicitly acknowledged. The conclusions are appropriately linked to the data and do not overreach.

Additional comments

The revision has addressed the key points from the initial review, particularly regarding methodological clarity, integration of figures, and transparency about limitations. The RTSB-C represents an important contribution by providing a culturally adapted, validated tool for assessing resistance training skills in Mandarin-speaking adolescents. This tool has clear relevance for researchers, educators, and policymakers, and the work aligns well with PeerJ’s aims in evidence-based health promotion.

Suggestions for further minor enhancement:
1. Expand the discussion of the potential ceiling effect in the lunge task.
2. Include a short statement on broader cultural considerations in motor skill execution and assessment in Chinese school settings.

·

Basic reporting

-

Experimental design

-

Validity of the findings

-

Additional comments

Dear authors,
The effort you have put into making corrections based on the reviewers' suggestions has significantly improved your paper, which is now a much more detailed, clearer, and more understandable representation of your research.
Congratulations!

---

## Round 0.3 · Minor Revisions

· Academic Editor

Minor Revisions

Thank you for revising your manuscript to address the remaining concerns of our reviewers. We are satisfied that their various comments have been satisfactorily addressed. The Section Editor for Sports Medicine and Rehabilitation, Mike Climstein, has nevertheless recommended some final revisions to enhance communication of your findings, which are available as tracked changes in the attached document. Please consider these changes as you revise your manuscript one last time then resubmit a final version for publication.

---

## Round 0.4 · accepted · Accept

· Academic Editor

Accept

Thank you for the final revisions to your manuscript, which is now ready for acceptance.